# Learning Heterogeneous Interaction Strengths by Trajectory Prediction with Graph Neural Network

**Seungwoong Ha**
Department of Physics
KAIST
Daejeon 34141, South Korea
{skyround2002}@gmail.com

**Hawoong Jeong**
Department of Physics
KAIST
Daejeon 34141, South Korea
{hjeong}@kaist.edu

## Abstract

Dynamical systems with interacting agents are universal in nature, commonly modeled by a graph of relationships between their constituents. Recently, various works have been presented to tackle the problem of inferring those relationships from the system trajectories via deep neural networks, but most of the studies assume binary or discrete types of interactions for simplicity. In the real world, the interaction kernels often involve continuous interaction strengths, which cannot be accurately approximated by discrete relations. In this work, we propose the relational attentive inference network (RAIN) to infer continuously weighted interaction graphs without any ground-truth interaction strengths. Our model employs a novel pairwise attention (PA) mechanism to refine the trajectory representations and a graph transformer to extract heterogeneous interaction weights for each pair of agents. We show that our RAIN model with the PA mechanism accurately infers continuous interaction strengths for simulated physical systems in an unsupervised manner. Further, RAIN with PA successfully predicts trajectories from motion capture data with an interpretable interaction graph, demonstrating the virtue of modeling unknown dynamics with continuous weights.

## 1 Introduction

Dynamical systems with interactions provide a fundamental model for a myriad of academic fields, yet finding out the form and strength of interactions remains an open problem due to its inherent degeneracy and complexity. Although it is crucial to identify the interaction graph of a complex system for understanding its dynamics, disentangling individual interactions from trajectory data without any ground-truth labels is a notoriously hard inverse problem. Further, if the interactions are heterogeneous and coupled with continuous strength constants, the interaction graph is called *weighted* and the inference became much harder with increased level of degeneracies.

In this work, we assume the dynamical system with $N$ objects (or agents), and their (discretized) trajectories $\mathbf{x}_1, \mathbf{x}_2, \ldots, \mathbf{x}_N$ from timestep $t = 0$ to $T$ are given. If the system has an interaction kernel $Q(\mathbf{x}_i, \mathbf{x}_j)$ and the dynamics are governed by a form of $\dot{\mathbf{x}}_{\mathbf{i}} = \sum_{j \neq i} k_{ij} Q(\mathbf{x}_i, \mathbf{x}_j)$ with some variable $k_{ij}$, which is prevalent in nature and physical system, we call $k_{ij}$ as an *interaction strength* between the object $i$ and $j$. With proper normalization, we can always regard $0 \leq k_{ij} \leq 1$. In general, $k_{ij}$ may have continuous values and forms a weighted interaction graph, which can be expressed in the form of a connectivity matrix $K$; a conventional adjacency matrix with continuous-valued entries of $k_{ij}$. Hence, the problem is inferring continuous adjacency matrix $K$ from trajectories $\mathbf{x}$ alone.

In the current work, we propose a neural network called Relational Attentive Inference Network (RAIN) to address the problem of inferring weighted interaction graphs from multivariate trajectory data in an unsupervised manner. RAIN infers the interaction strength between two agents from previous trajectories by learning the attentive weight while simultaneously learning the unknown dynamics of the system and thus is able to precisely predict the future trajectories. Our model employs the attention mechanism twice: once for the construction of pairwise trajectory embedding

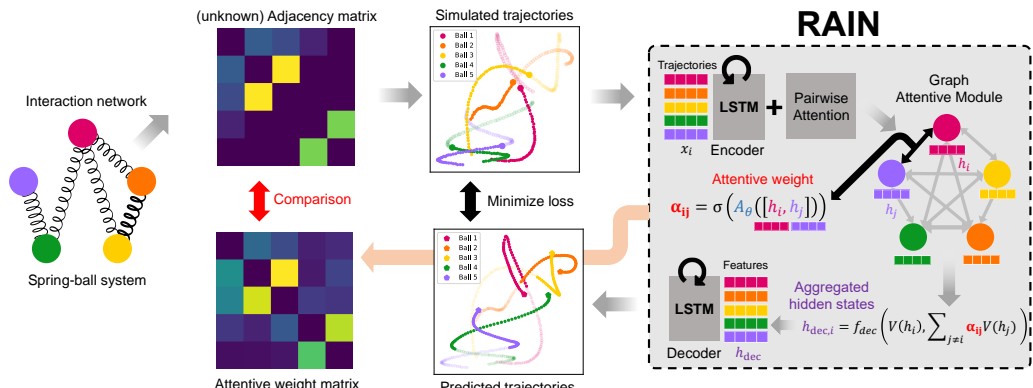

Figure 1: Overview of system formulation and RAIN architecture. RAIN encodes each agent's trajectory with an LSTM encoder and applies pairwise attention (PA) to the hidden states for constructing a pair of embeddings for each agent pair. Then the graph attentive module extracts the interaction strength from a pair of embeddings in the form of an attention weight with an MLP, $A_\theta$. The decoder module finally predicts the future trajectories of each agent with an LSTM decoder, but here, each prediction can only employ the weighted information from other agents. This restriction on information induces the attention weights in the learning process to properly reflect the strengths of the connections.

and once for the actual graph weight extraction. Differing from previous approaches such as the graph attention network (GAT) Veličković et al. (2017), RAIN aims to infer the absolute interaction strength that governs the system dynamics by employing attention module with multilayer perceptron (MLP) and sigmoid activation. By comparing the inferred interaction strengths of simulated physical systems with ground-truth values that are not provided at the training stage, we verify that RAIN is capable of inferring both system dynamics and weighted interaction graphs solely from multivariate data. We further show that RAIN outperforms discrete baselines on real-world motion capture data, representing a system in which we cannot be certain whether a continuous form of interaction strengths even exists. In this way, we demonstrate that the rich flexibility and expressibility of the continuous modeling of interaction strengths are crucial for the accurate prediction of the future dynamics of an unknown empirical system.

## 2 RELATED STUDIES

There has been a long history and substantial amount of work on both inferring the network topology and the nonlinear correlation between interacting constituents from data Casadiego et al. (2017); Ching & Tam (2017); Chang et al. (2018); Shi et al. (2020); Ha & Jeong (2021); Fujii et al. (2021), along with the development of various measures to capture the relation between constituents (e.g., Pearson correlations, mutual information, transfer entropy, Granger causality, and variants thereof Schreiber (2000)). Many of these inferences focus on specific systems with the necessity for a model prior, such as domain knowledge of the agent characteristics, proper basis construction, and detailed assumptions on the system dynamics.

Recently, by phenomenal advances in machine learning, adopting a neural network as a key component of the interaction inference has gained attention from researchers Veličković et al. (2017); Kipf et al. (2018); Zhang et al. (2019). The key strength of these approaches comes from the fact that a neural network enables relatively free-form modeling of the system. One influential work in this direction, neural relational inference (NRI) Kipf et al. (2018), explicitly infers edges by predicting the future trajectories of the given system. But previous studies for extracting interaction graphs with neural networks Kipf et al. (2018); Webb et al. (2019); Graber & Schwing (2020) mainly focused on inferring edges with discrete edge types, which means that they are incapable of distinguishing the edges of different interaction strengths with the same type. Considering the common occurrence of such heterogeneous interaction strengths throughout diverse systems, the assumption of discrete edge types severely limits the expressibility of the model.

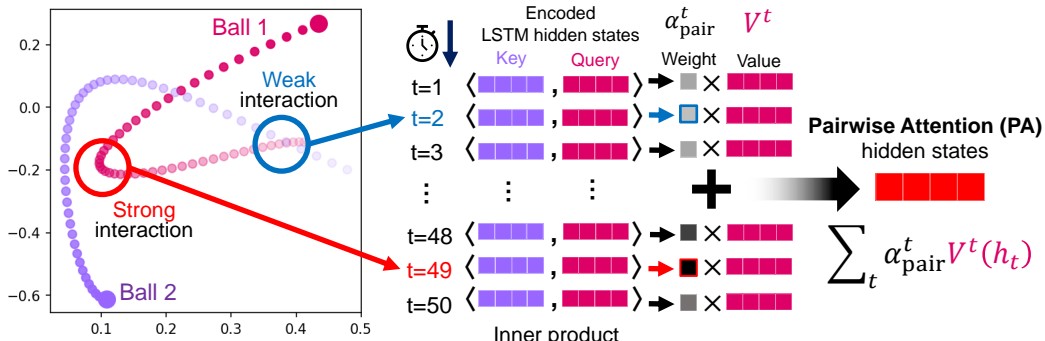

Figure 2: Overview of the PA mechanism, which facilitates effective interaction inference by emphasizing the critical part of the time series. Here, two balls weakly interacted at first but showed strong interaction at the end. The circled points at timestep 2 and 49 are emphasized as exemplary timestep where the interaction was weak and strong, respectively. PA achieves refined trajectory representations by comparing encoded LSTM hidden states at the same time. Due to the asymmetric nature of the transformer architecture, the weighted hidden state for A to B and B to A can be drastically different, which lets PA properly handle the directional asymmetry of interaction.

One recent study Li et al. (2022) tackles this problem of inferring the connectivity matrix by first training the neural network and then additionally applying a 'graph translator' to extract continuous graph properties, which needs a ground-truth label to train. Several studies from physicists Zhang et al. (2015); Lai (2017) employed perturbation analysis and response dynamics to infer continuous interaction strengths. However, these correlation-based methods usually require more than $10^5$ to $10^7$ data points for the inference, which is a feasible size for an entire dataset but not for a single instance, and thus the direct application to experimental data is difficult.

Also, many of the recent studies in relational inference Li et al. (2020; 2021); Sun et al. (2022) and trajectory prediction Xu et al. (2022b); Mo et al. (2022); Xu et al. (2022a) explicitly model the relationship between objects as a continuous value. But, for interaction inference, all these models rely on the traditional Graph Attention (GAT) Veličković et al. (2017) or slightly modified version of it, which becomes a limitation for challenging inference problem. Further, most of these works are not focusing on correctly infer the true interaction strength, hence only tested on the system with unknown ground-truth interaction strength or didn't report any comparison with it. In this work, we show that even the advanced and strictly more expressive version of GAT, GATv2 Brody et al. (2021) is insufficient to fully capture the true interaction strength.

## 3 MODEL DESCRIPTION

Our RAIN model as shown in Fig. 1 comprises three parts trained jointly: an encoder that compresses time series data, a graph extraction module that infers the interaction weight between every pair of agents, and a decoder that predicts the future trajectories of each agent. Note that RAIN does not require a ground-truth interaction graph for training; instead, it produces an interaction graph as a byproduct of future trajectory prediction. In the following, we formalize our model and describe each component in detail.

### 3.1 ENCODER AND PAIRWISE ATTENTION

The long short-term memory (LSTM) encoder is composed of a single layer of gated recurrent unit (GRU) Cho et al. (2014) with hidden state dimension $d_{\text{lstm}} = 128$. The encoder repeatedly takes the trajectory data of all agents from each timestep by receiving $T_{\text{enc}}$ steps of trajectories $\mathbf{x}^1, \mathbf{x}^2, \ldots, \mathbf{x}^{T_{\text{enc}}}$, each consisting of $R$ state variables of $N$ agents at time $T$, and producing corresponding $T_{\text{enc}}$ hidden states, $\mathbf{h}^1, \mathbf{h}^2, \ldots, \mathbf{h}^{T_{\text{enc}}}$. The last hidden state $\mathbf{h}^{T_{\text{enc}}}$ could preserve the information from the entire trajectory of all agents in theory. But we found that a naive final hidden state is insufficient to fully capture the interaction strength in large-scale systems. In this study, we

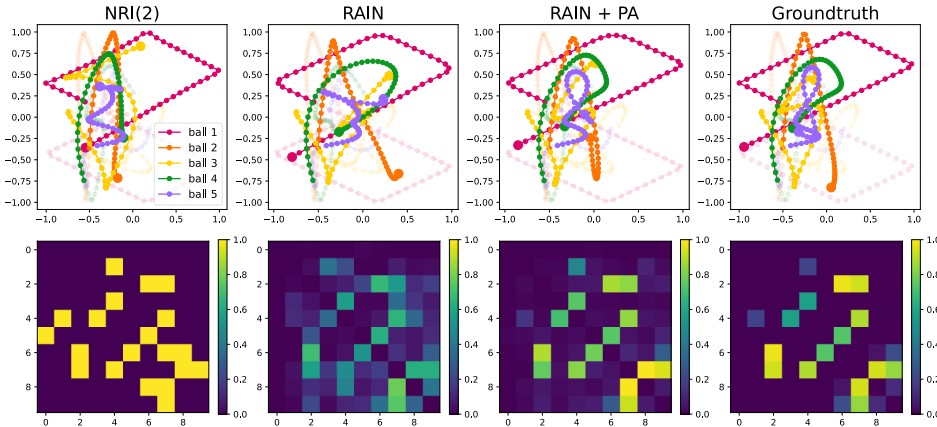

Figure 3: Visualization of the trajectory predictions (upper) and retrieved connectivity matrices (lower) for the spring-ball system. The results from left to right are from NRI(2), RAIN without PA, RAIN with PA, and ground-truth. Here, 5 out of 10 balls are drawn for trajectory visualization, and semi-transparent paths indicate the first 50 steps of input trajectories while solid paths denote 50 steps of predicted future trajectories.

propose a *pairwise attention* (PA) mechanism to effectively infer the interaction strength between two trajectories. The intuition behind PA is straightforward: a single hidden state cannot be a suitable choice for extracting the interaction strengths from *every* possible pair. Thus, we calculate the attention between same-time hidden states to assign weights to their contribution, as depicted in Fig. 2. Here, the term 'same-time' in PA indicates that the attention values are calculated between $h_i^t$ and $h_j^t$ ($K_{\text{pair},i}^t$ and $Q_{\text{pair},j}^t$, technically) only for same time t, not between every possible $t = 0\,T$, as conventional full attention mechanism does. We use a slightly modified transformer Vaswani et al. (2017) with $m = 4$ heads and a head dimension of $d_{\text{h}} = d_{\text{lstm}/m=32}$ to emphasize strong interaction. See Vaswani et al. (2017) for a detailed description of the transformer architecture. Formally, the LSTM hidden states are processed into Key($K$), Query($Q$), and Value($V$) matrices as follows,

$$\mathbf{h}_i^t = f_{\text{LSTM}}(\mathbf{h}_i^{t-1}, \mathbf{x}_i^t) \tag{1}$$

$$X_{\text{pair},i}^t = f_{\text{pair},X}(\mathbf{h}_i^t) \quad \text{where } X \in \{K, Q, V\} \tag{2}$$

$$X_{\text{pair},i}^t = X_{\text{pair},i}^{t,1} \oplus X_{\text{pair},i}^{t,2} \oplus \ldots \oplus X_{\text{pair},i}^{t,m}, \tag{3}$$

where the symbol $\oplus$ indicates the concatenation of the matrices at the last dimension, $f_{\text{LSTM}}$ is the GRU layer, $\mathbf{x}_i^t$ are the state variables of the $i$th agent at time $t$, and $f_{\text{pair},X}$ is a stack of MLP layers for $K_{\text{pair}}$, $Q_{\text{pair}}$, and $V_{\text{pair}}$. The superscripts $t$ and $m$ on $X$ indicate the time and head number, respectively. The attention-weighted hidden state $\tilde{\mathbf{h}}$ is expressed as follows,

$$\alpha_{\text{pair},ij}^{t,m} = \text{softmax}(K_{\text{pair},i}^{t,m}(Q_{\text{pair},j}^{t,m})^T / \sqrt{d_{\text{h}}}) \tag{4}$$

$$\tilde{\mathbf{h}}_i^m = \sum_{t=1}^{T_{\text{enc}}} \alpha_{\text{pair},ij}^{t,m} V_j^{t,m} \tag{5}$$

$$\tilde{\mathbf{h}}_i = \tilde{\mathbf{h}}_i^1 \oplus \tilde{\mathbf{h}}_i^2 \oplus \ldots \oplus \tilde{\mathbf{h}}_i^m, \tag{6}$$

where $\alpha_{\text{pair},ij}^{t,m}$ is an attentive weight between agents $i$ and $j$, and $\tilde{\mathbf{h}}_i^m$ is a weighted hidden state for the $m$th attention head. RAIN passes $\tilde{\mathbf{h}}$ to the graph extraction module.

By employing the PA mechanism, we can refine the LSTM hidden states by focusing on the specific time of interaction, which is unique to each agent pair. We highlight that comparing the same-time hidden states only, rather than considering the full attention as in conventional settings, significantly

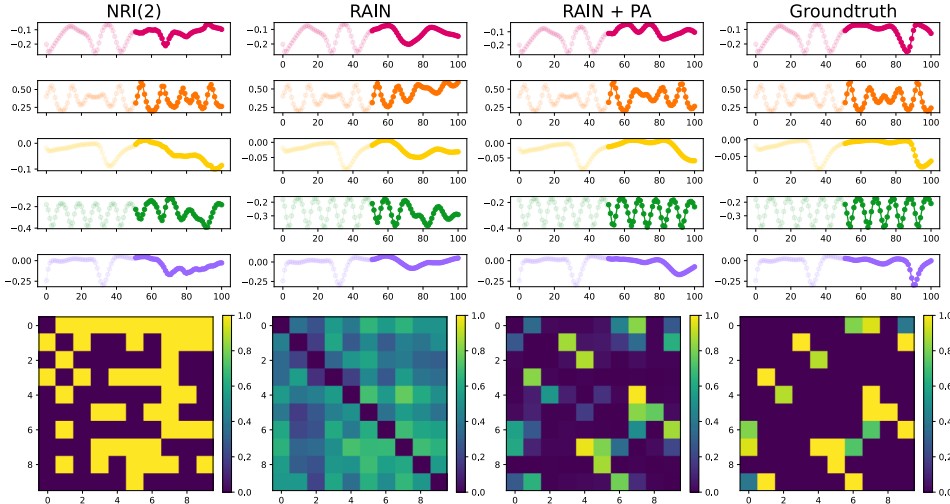

Figure 4: Visualization of the trajectory predictions (upper) and retrieved connectivity matrices (lower) for coupled Kuramoto oscillators. The results from left to right are from NRI(2), RAIN without PA, RAIN with PA, and ground-truth. Here, 5 out of 10 oscillators are drawn for trajectory ($\frac{d\phi_i}{dt}$) visualization, and semi-transparent paths indicate the first 50 steps of input trajectories while solid paths denote 50 steps of predicted future trajectories.

reduces the time complexity of the inference while achieving the goal of the extraction of temporal information. Also, since the transformer can handle asymmetric relationships by differentiating key and query, PA can capture directional connections with ease.

## 3.2 GRAPH EXTRACTION MODULE

The outline of the graph extraction module of RAIN is similar to the GAT Veličković et al. (2017). The difference between ours and the original GAT is that RAIN has no prior knowledge of the underlying graph and thus the $\alpha_{\text{graph}}$ attention value should infer the presence of the edge itself as well as its strength, while GAT aims to find the relative importance between fixed graph edges. Also, the conventional inner-product attention of GAT is replaced with a 3-layer MLP $A_\theta$ in RAIN to achieve much more flexible representations. We use attention-weighted hidden states from the PA mechanism to apply the transformer as $\alpha_{\text{graph},ij} = \sigma(A_\theta(\tilde{\mathbf{h}}_i \oplus \tilde{\mathbf{h}}_i))$. Note that instead of softmax which normalizes the attention weight, sigmoid activation $\sigma$ is used for graph attention to obtain the *absolute* interaction strength. This is because our main goal is to infer the ground-truth interaction strength, not the relative strength for a single instance. Extracted graph attention $\alpha_{\text{graph},ij}$ becomes a weight for the decoder module.

## 3.3 DECODER

The decoder shares the same GRU layer with the encoder module and the employ new value function, $V_{\text{dec},i}$, for the message aggregation. For agent $i$, RAIN concatenates attention-weighted value vectors from other agents along with its own value vector as follows,

$$\mathbf{h}_i^{T_{\text{enc}}+t} = f_{\text{LSTM}}(\mathbf{h}_i^{T_{\text{enc}}+t-1}, \mathbf{x}_i^{T_{\text{enc}}+t}) \tag{7}$$

$$V_{\text{dec},i}^{T_{\text{enc}}+t} = f_{\text{pair},V}(\mathbf{h}_i^{T_{\text{enc}}+t}) \tag{8}$$

$$\tilde{\mathbf{h}}_{\text{dec},i}^{T_{\text{enc}}+t} = \sum_{j \neq i} \alpha_{\text{graph}} V_{\text{dec},j}^{T_{\text{enc}}+t} \oplus V_{\text{dec},i}^{T_{\text{enc}}+t}, \tag{9}$$

where $f_{\text{LSTM}}$ and $f_{\text{pair},V}$ consists of MLP. RAIN finally produces a prediction of the state variables of the next step using another stack of MLP layers $f_{\text{dec}}$, $f_\mu$, and $f_\sigma$ by first $\mathbf{h}_{\text{dec}}^{T_{\text{enc}}+t} = f_{\text{dec}}(\tilde{\mathbf{h}}_{\text{dec},i}^{T_{\text{enc}}+t})$, and

then $\mu_i^{T_{enc}+t} = f_\mu(\mathbf{h}_{dec}^{T_{enc}+t})$ and $\sigma_i^{T_{enc}+t} = f_\sigma(\mathbf{h}_{dec}^{T_{enc}+t})$. The outputs of $f_\mu$ and $f_\sigma$ are $r$-dimensional vectors, each representing the means and variances of the difference of state variables. RAIN samples the next state from a Gaussian distribution with $f_\mu$ and $f_\sigma$ and adds the values to the previous state. The decoder iterates this for $T_{dec}$ steps to predict the future states. For training, we employ negative log-likelihood (NLL) loss for a Gaussian distribution.

## 4 EXPERIMENTS

We demonstrate the capability of RAIN by performing inference tasks with various model systems ranging from simulated physical systems to real motion capture data from a walking human. All of the models and baselines are implemented with `PyTorch` and optimized with Adam Kingma & Ba (2014). Data and models are available at github.com/nokpil/RAIN. See the Supplementary Materials for details on the data generation, model architectures, and training details.

### 4.1 PHYSICAL SIMULATIONS

We simulated the trajectories of two physical systems: a spring-ball system and phase-coupled Kuramoto oscillators Kuramoto (1975). Unlike previous studies Kipf et al. (2018); Webb et al. (2019) in which every interaction strength $w$ in the system is discrete and constant (commonly $w = 1$), we consider a more general setting where interaction strengths are drawn from a continuum and are thus heterogeneous. Assuming that two agents $i$ and $j$ are interacting, the interaction strength for each system would be a spring constant $k_{ij}$ for a spring-ball system and a coupling weight $w_{ij}$ for a Kuramoto model. Although our model can handle asymmetric interaction strength—thanks to the asymmetric nature of the transformer—we take the symmetric form of connectivity matrix $k_{ij} = k_{ji}$ for the simulated systems. For the spring-ball and Kuramoto systems, we first select the edges between $n$ nodes with probability $p$ (excluding self-connections) to construct an interaction graph, and then assign a randomly sampled interaction strength to each edge from a uniform distribution $U[0, 1]$ while expressing non-assigned edges with zero interaction strength. We generate 10k training samples and 2k validation samples for all simulated tasks. The state variables consisting of trajectories are $x, y, v_x, v_y$ (positions and velocities) for the spring-ball system. For the Kuramoto oscillators, a concatenated vector of $\frac{d\phi}{dt}$, $\sin\phi$, and intrinsic frequency $\omega$ are used to construct the trajectories, where $\phi$ is the phase of an oscillator.

For evaluation, we measure how accurately the model predicts future trajectories by mean squared error (MSE) and how well the model retrieves the original connectivity matrix by two forms of Pearson correlation. We first gather every retrieved interaction strength $a$ and corresponding true interaction strength $k$ from all 2k validation samples and calculate the correlation in total ($\rho_{tot}$). Secondly, we independently calculate the correlation for each validation sample and take the average of the 2k samples ($\rho_{sample}$). One may interpret $\rho_{tot}$ as the overall performance of the model, while $\rho_{sample}$ indicates the expected correlation for a single instance at a test time. We excluded diagonal trivial zeroes (due to no self-loop interaction) while calculating correlation, so the reported value is strictly lower compared to the case where every value in the connectivity matrix is used. Correlations from the baseline models with discrete edge types are obtained by assigning continuous weights to each edge type according to every possible permutation and choosing the best one. More precisely, we assign $n$ weights of $0, \frac{1}{n-1}, \frac{2}{n-1}, \dots, 1$ to $n$ edge types with $n!$ different assignments and report the highest correlation.

### 4.1.1 RESULTS

Table 1 shows the correlations between the true interaction graph and the predicted interaction strengths from various models. Following Kipf et al. (2018), we measure the correlation between trajectory feature vectors as Corr. (Path) and the correlation between trained LSTM feature vectors as Corr. (LSTM). NRI(2) and NRI(4) indicate the NRI from Kipf et al. (2018) with 2 and 4 edges types, respectively. As a baseline with continuous attention, we employ GATv2 Brody et al. (2021), which is an advanced version of GAT and analytically proven to be strictly more expressive than the original GAT. We check the effect of PA by testing the RAIN model both with and without the PA mechanism. In Table 1, we can clearly see that RAIN with PA significantly outperforms every other model at inferring interaction strengths. Here, GATv2 showed worse performance in the

Table 1: Total Pearson correlation ($\rho_{\text{tot}}$) and the sample Pearson correlation ($\rho_{\text{sample}}$) between retrieved interaction weights and true interaction weights for simulations with 5 and 10 interacting objects.

| Model | Spring | | Kuramoto | | Spring | | Kuramoto | |
|---|---|---|---|---|---|---|---|---|
| Corr. | $\rho_{\text{tot}}$ | $\rho_{\text{sample}}$ | $\rho_{\text{tot}}$ | $\rho_{\text{sample}}$ | $\rho_{\text{tot}}$ | $\rho_{\text{sample}}$ | $\rho_{\text{tot}}$ | $\rho_{\text{sample}}$ |
| | 5 objects | | | | 10 objects | | | |
| Corr. (Path) | 0.2917 | 0.2944 | 0.0243 | -0.1206 | 0.2654 | 0.2732 | 0.1047 | 0.0682 |
| Corr. (LSTM) | 0.0979 | 0.0952 | -0.1369 | -0.0751 | 0.1196 | 0.1096 | 0.0024 | -0.0012 |
| NRI(2) | 0.8482 | 0.8660 | 0.0027 | -0.0048 | 0.8602 | 0.8593 | 0.0393 | 0.0413 |
| NRI(4) | 0.9250 | 0.9214 | 0.0230 | 0.0192 | 0.8921 | 0.8966 | 0.0438 | 0.0454 |
| GATv2 | 0.8425 | 0.8643 | 0.6296 | 0.6466 | 0.7981 | 0.8060 | 0.5410 | 0.5481 |
| RAIN | 0.8411 | 0.8770 | 0.5213 | 0.4987 | 0.7787 | 0.8164 | 0.4560 | 0.4683 |
| RAIN + PA | **0.9400** | **0.9568** | **0.8731** | **0.8925** | **0.9117** | **0.9221** | **0.8265** | **0.8411** |

Table 2: Mean squared error (MSE) in predicting future states for simulations with 5 and 10 interacting objects. Underlined entries show better results than those from RAIN with PA.

| Model | Spring | | | Kuramoto | | |
|---|---|---|---|---|---|---|
| Prediction steps | 10 | 30 | 50 | 10 | 30 | 50 |
| | 5 objects | | | | | |
| Static | 0.6665 | 1.3614 | 1.2340 | 1.0671 | 0.9901 | 1.0108 |
| SingleLSTM | 0.1969 | 0.5643 | 0.6048 | 0.5006 | 0.4957 | 0.5247 |
| JointLSTM | 0.0704 | 0.3913 | 0.6362 | 0.0391 | 0.1083 | 0.1749 |
| NRI(2) | 0.0171 | 0.2232 | 0.5263 | 0.0268 | 0.1615 | 0.3275 |
| NRI(4) | 0.0122 | 0.1395 | 0.3573 | 0.0308 | 0.1498 | 0.3050 |
| GATv2 | 0.0230 | 0.2081 | 0.4252 | 0.0455 | 0.1901 | 0.3140 |
| RAIN | 0.0259 | 0.1322 | 0.3113 | 0.0388 | 0.1705 | 0.3675 |
| RAIN + PA | **0.0084** | **0.0714** | **0.2193** | **0.0041** | **0.0645** | **0.2059** |
| RAIN (true graph) | 0.0062 | 0.0181 | 0.0732 | 0.0037 | 0.0068 | 0.0122 |
| | 10 objects | | | | | |
| Static | 0.6070 | 1.1253 | 1.0779 | 1.0380 | 0.9935 | 0.9799 |
| SingleLSTM | 0.2028 | 0.4991 | 0.5189 | 0.5511 | 0.5138 | 0.5173 |
| JointLSTM | 0.1317 | 0.4823 | 0.5840 | 0.0953 | 0.2490 | 0.3832 |
| NRI(2) | 0.0078 | 0.1158 | 0.3169 | 0.0392 | 0.2341 | 0.4451 |
| NRI(4) | 0.0061 | 0.0866 | 0.2440 | 0.0372 | 0.2411 | 0.4385 |
| GATv2 | 0.1309 | 0.2634 | 0.4192 | 0.0354 | 0.1665 | 0.3766 |
| RAIN | 0.1665 | 0.3109 | 0.4996 | 0.0307 | 0.1902 | 0.3942 |
| RAIN + PA | **0.0069** | **0.0892** | **0.2351** | **0.0115** | **0.1586** | **0.4016** |
| RAIN (true graph) | 0.0059 | 0.0163 | 0.0504 | 0.0009 | 0.0063 | 0.0301 |

spring-ball system and better performance in Kuramoto oscillators compared to RAIN, but significantly worse than RAIN with PA model in both cases. Also, note that the high correlations from the NRI models for the spring-ball system can only be achieved if and only if we know the ground-truth interaction strength and choose the best permutation. Considering the huge performance difference between RAIN without and with PA, our model and PA mechanism are both crucial for accurate interaction strength retrieval. We show MSE results for predicting 50 future states in Table 2, Here, again following Kipf et al. (2018), SingleLSTM runs a single LSTM for each object separately, while JointLSTM concatenates all state vectors (thus may handle a fixed number of agents only) and trains a single LSTM to jointly predict all future states. See the Supplementary Materials for further analysis, including full histograms of the correlation distributions.

Figures 3 and 4 show the results of future prediction and interaction strength inference for 10 interacting agents, where 5 trajectories are drawn for visualization. In Fig. 3, we can observe that all

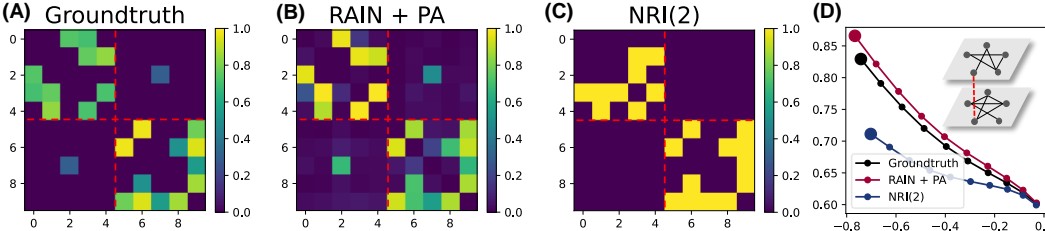

Figure 5: (A) Ground-truth connectivity matrix of the given multilayer network with a single inter-layer link. Inferred interaction weights from (B) RAIN with PA and (C) NRI(2) are shown. (D) Trajectory predictions (10 steps) for ball 3 on the $xy$-plane. The inset depicts a diagram of the given multilayer networks, where the inter-layer link is highlighted with the red dotted line.

models, including NRI(2), succeed in capturing the existence of interaction between agents. But it is apparent that the NRI(2) model fails to capture the interactions with small weights, while RAIN without PA yields in numerous false positives with spurious weights. Such weakness of each model is reflected in relatively large prediction errors compared to RAIN with PA, especially for ball 4 (green) and ball 5 (purple). The power of the PA mechanism becomes more evident with the Kuramoto oscillators (Fig. 4), where only RAIN with PA succeeds in retrieving meaningful interaction weights. To sum up, we can conclude that the PA mechanism clearly helps the refinement of hidden states and thus yields better results with a greater correlation with the ground-truth interaction weights.

### 4.1.2 IMPACT OF WEAK INTERACTION

By inferring interaction with a continuous strength as RAIN does, we can capture an entire spectrum of interactions with a single model. Particularly, we find that our model is able to detect weak interactions that are often ignored by the discrete NRI models due to their limited capacity. In Figure 5, we emphasize the significance of the inference of weak interaction by constructing the connectivity matrix in a form of a multilayer network Kivelä et al. (2014) with 2 layers. Here, we prepare two densely connected layers of springs where their spring constants are sampled from $[0.5, 1]$, uniformly. Between the two layers, we set a single inter-layer link (connecting ball 3 and ball 8) with a spring constant of $0.3$, which is much smaller than the intra-layer interaction weights. Since the synchronization of a multilayer network largely depends on the strength and structure of inter-layer coupling Leyva et al. (2017); Della Rossa et al. (2020), capturing this weak but significant connection between the two layers is critical for trajectory prediction. As Fig. 5 shows, RAIN with PA accurately captures this weak interaction, while NRI(2) misses it and thus its predicted trajectory considerably deviates from the true one.

### 4.2 MOTION CAPTURE DATA

Next, we test our model in inferring interactions between the joints of a walking human data from the CMU motion database cmu (2003). Different from the physical simulations, this real-world system does not possess any well-known dynamics with a continuous interaction strength. Following Kipf et al. (2018) and Graber & Schwing (2020), we use the data from subject #35 with 31 joints. We split the training and validation data sets into a $4:1$ ratio. We use the same protocol as with the physical systems: provide $50$ time steps of trajectories and let the model predict $50$ unseen time steps afterward. For the motion data, we use raw value of the future state instead of the difference from the current state to calculate the Gaussian loss for fast convergence.

Figure 6A shows the errors from each model's prediction, where RAIN with PA achieved the lowest MSE. An example visualization of the prediction is shown in Fig. 6B. Without dynamically re-evaluating the interaction graph at every time step as reported in Kipf et al. (2018) and Graber & Schwing (2020), our model precisely predicts the future states from static but continuous interaction weights. Also, as shown in Fig. 6C, our model produces a smooth and block-wise connectivity matrix which is much easier to interpret compared to that from the discrete NRI(2). This is because there is no restriction or natural meaning for each edge type in the discrete model, and thus the

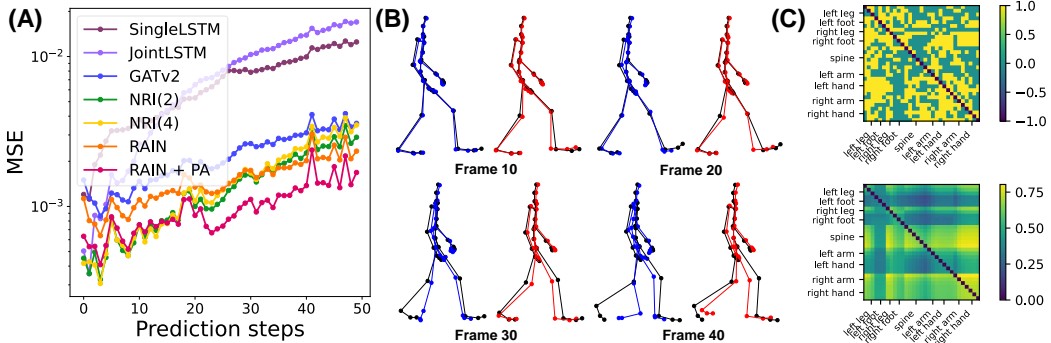

Figure 6: (A) Validation MSE comparison for motion capture data. The prediction results of 50 time steps from SingleLSTM, JointLSTM, NRI(2), NRI(4), RAIN without PA, and RAIN with PA are shown. (B) Sample predictions from NRI(2) (blue) and RAIN with PA (red) for a validation trajectory of motion capture data with the ground-truth states (black). (C) Inferred interaction weights from NRI(2) (top) and RAIN with PA (bottom) are shown. Here, the diagonal entries from NRI(2) are marked with $-1$ to distinguish them from the two edge types, 0 and 1.

manual sparsity prior is needed to handle the 'no interaction' edges if an abundance is expected. On the other hand, the attention value of RAIN is expected to convey the strength of interaction directly. For instance, it is clear from RAIN's connectivity matrix that the dynamics of the feet are less correlated with the rest of the joints and also that the spine moves along with the right leg, arm, and hand, both of which agree with the qualitative movements of a human while walking.

## 5 CONCLUSION

In this work, we introduced RAIN, a model for inferring continuous weighted interaction graphs from trajectory data in an unsupervised manner. With the PA mechanism that computes the attention between same-time LSTM hidden states between agents, we can sharply refine the representation for the interaction weight extraction. Our model successfully inferred the absolute interaction weights from simulated physical systems and further showed great prediction performance with an empirical system, demonstrating the advantage of continuous weight modeling in relational learning. Notably, RAIN needs only 50 time steps of data to infer the interaction weights, and thus requires less data by several orders of magnitude than correlation-based theoretical methods Zhang et al. (2015); Lai (2017).

Also, we found that the refinement of the LSTM hidden state with PA is critical for meaningful performance. Since the PA mechanism is generally applicable to a neural model where its trajectory is represented by a recurrent neural network, one may expect an increase in performance by employing PA in other relational models without increasing the inference time significantly. If the system size $N$ is extremely large and $O(N^2)$ time complexity of the attention mechanism becomes a bottleneck, one may adopt recently proposed attention mechanisms with linear time complexity and virtually no performance drop Wang et al. (2020); Shen et al. (2021) to calculate PA.

In the real world, interactions between agents in complex systems possess a broad range of characteristics. For instance, they can be either positive (excitatory or encouraging) or negative (inhibitory or suppressing), time-delayed with heterogeneous time scales, and noisy both inherently and externally. Complex systems in nature generally contain every aspect of these characteristics, such as neural signals in a human brain. Reinforcing the current RAIN architecture to handle data of such complex nature with a single model would be a promising future direction to explore.

## 6 DATA AVAILABILITY

All of the codes for data generations and models are available at github.com/nokpil/RAIN.

## 7 ACKNOWLEDGEMENT

This research was supported by the Basic Science Research Program through the National Research Foundation of Korea NRF-2022R1A2B5B02001752 (HJ) and National Research Foundation of Korea Grant funded by the Korean Government NRF-2018S1A3A2075175 (SH).

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
