# OpenReview forum: "Learning Heterogeneous Interaction Strengths by Trajectory Prediction with Graph Neural Network"
_ICLR.cc/2023/Conference — ICLR 2023 poster_

### Official Review · Reviewer_jYRz · 2022-10-13

**Confidence:** 3
**Correctness:** 3
**Technical Novelty And Significance:** 2
**Empirical Novelty And Significance:** 3
**Recommendation:** 6

**Clarity, Quality, Novelty And Reproducibility:**

- The presentation is OK, but I think there are typos in their equations.

- There are not enough existing works discussed in this paper.

**Strength And Weaknesses:**

--- Strengths

- This paper presents a sound method for predicting agent interaction and future trajectory. The proposed method achieves better performance than the baseline method on public benchmarks.

- The result shows that adding pair-wise attention helps.


--- Weaknesses

- There is not sufficient comparison against the state-of-the-art baselines to demonstrate the advantage of the proposed method. In the evaluation section, NRI is the only public baseline that the authors compared their method against, and it's quite an old baseline. Also, there is not much discussion about the existing works in the paper.


--- Other comments

- I think there are some typos in Eq (5). It should be V_j^{t,m} instead of V_i^{t,m}. Also, I think there should be a sum over j. Otherwise, why does the subscript j disappear in the left-hand side of the equation?

- I suggest the authors adjust the positions of the result tables and figures. For example, Table 1 and Table 2 are referenced in the same paragraph in the text, but one is on Page 6 and one is on Page 9.

**Summary Of The Paper:**

This paper presents a model called RAIN for interaction and trajectory prediction. Given the history trajectories of multiple moving agents, the RAIN model predicts their interaction weights and future trajectories.

The RAIN model works as follows. 1) It first encodes the history of each agent with an LSTM encoder. 2) It then uses pairwise attention to enrich the per-agent feature vectors. 3) It then uses a graph attention network to further infer the interaction weights. 4) Finally, it uses an LSTM decoder to predict their future trajectories.

The authors evaluated the RAIN model on two physical systems (a spring-ball system and a phase-coupled Kuramoto oscillator system) and a human walking dataset. They used the NRI method as their baseline. The evaluation result shows that the RAIN model significantly increases the accuracy of correlation prediction and reduces the trajectory MSE.

**Summary Of The Review:**

This paper presents a sound method for predicting agent interaction and future trajectory. The proposed method achieves better performance than the baseline method on public benchmarks. So I am giving it a weak accept.

---

> ### Author Response · Authors · 2022-11-15
> **Response to jYRz**
>
> Thank you for your thorough and detailed review of the manuscript. We carefully checked the issues and revised our manuscript as follows. (We recommend you check the overall comment before reading this.)
>
> ---
>
> 1. As mentioned in the overall comment, we added a recent baseline, GATv2 (2021), to enrich our performance comparison and verified our model's superiorities over the new baseline. (will be ready by 11/17) Also, we separated the 'Related studies' section from the current introduction and added more recent references to help the reader's understanding of the field.
>
> ---
>
> 2. As you noticed, it should be the sum over j, and we corrected the typo. Thank you for finding such a detailed but critical mistake.
>
> ---
>
> 3. We are aware of the formatting issues and revised the positioning of the tables to match the main text in the current version.
>
> ---
>
> We hope our revised manuscript and responses have helped you understand our work better, and we look forward to your positive feedback. Thank you!

---

> > ### Comment · Reviewer_jYRz · 2022-11-17
> > **Thank you for your response**
> >
> > Thank you for your response. I will keep my score.

---

### Official Review · Reviewer_cYy1 · 2022-10-23

**Confidence:** 5
**Correctness:** 3
**Technical Novelty And Significance:** 3
**Empirical Novelty And Significance:** 2
**Recommendation:** 6

**Clarity, Quality, Novelty And Reproducibility:**

Details are in [Weaknesses].

**Clarity**:
The clarity is good but with some minor mistakes.

**Quality**:
Clear-written and well-organized.

**Novelty**:
Motivation is good and novelty are good as well.

**Reproducibility**:
The training procedure is clear,  and authors also provide the code, even though I searched to have.

**Strength And Weaknesses:**

**Strengths**:
1. Well-organized and clearly written.
2. Strong motivation with promising results.


**Weaknesses**:
1. In Section 3.1, when evaluating on the simulated physical systems, the procedure of obtaining the ground-truth of connectivity/interaction matrix is not clear, since the method in the unsupervised manner. Why use a single inter layer to get the "ground truth" matrix? More details and explanations should be included.


2. In Section 3.2, when it comes to the motion prediction problem, especifically using CMU motion dataset, existing methods usually predict the difference from the previous time step, but authors predict the raw value here. What's the reasons/insights here?  Also, in Figure 6(c), are there any ground-truth interaction weights? or how to tell which one is better when comparing the interaction weight matrix?


3. In Section 2.1, for the model complexity, for the pair-wise attention weights calculation, at each time step, it would be $O(N^{2})$, assuming there are $N$ objects. Also, when employing the transformer structure, the calculation cost would be very large. More explanations about reducing  time complexity should be included, some experiments would be better to illustrate.


4. Experiments are not sufficient, only NRI is compared (2018), more existing SOTA baselines should be included.

5. Missing related works, it is not a mandatory task, but I strong encourage authors to include a related work section to introduce some recent works on, i.e., interaction inference/learning[1,2], trajectory/motion prediction [3, 4], etc, and to highlight the differences between the proposed method and existing works.


6. Minor mistakes:
(1) In Figure 2, is it "+" or "$\bigoplus$"?
(2) In Figure 5 (D), x/y axis label is missing.
(3) In References, the names of conferences or journals, i.e., format and first letter, are not consistent.
I highly encourage authors to revise the submission to avoid such small mistakes, as well as some typos.

[1] GRIN: Generative Relation and Intention Network for Multi-agent Trajectory Prediction. NeurIPS2021
[2] Evolvegraph: Multi-Agent Trajectory Prediction with Dynamic Relational Reasoning. NeurIPS2020
[3] Adaptive Trajectory Prediction via Transferable GNN. CVPR2022
[4] GroupNet: Multiscale Hypergraph Neural Networks for Trajectory Prediction With Relational Reasoning. CVPR2022

**Summary Of The Paper:**

In this paper, authors propose a relational attentive inference network (RAIN), in which the interactions between agents are continuous. A pairwise attention (PA) is adopted to refine the representations and a transform structure is also employed to extract the interaction weights. The proposed method achieves good performance on interaction strength inference and promising results on motion prediction.

**Summary Of The Review:**

I list my concerns in [Weaknesses], I am happy to discuss and increase the rating if my concerns are addressed.

---

> ### Author Response · Authors · 2022-11-15
> **Response to Reviewer cYy1 (1/2)**
>
> Thank you for your thorough and detailed review of the manuscript. We carefully checked the issues and revised our manuscript as follows. (We recommend you check the overall comment before reading this.)
>
> ---
>
> 1. As mentioned in the overall comment, we added a recent baseline, GATv2 (2021), to enrich our performance comparison and verified our model's superiorities over the new baseline. (will be ready before 11/17) Also, we separated the 'Related studies' section from the current introduction and added more recent references to help the reader's understanding of the field.
>
> ---
>
> 2. Here is our attempt to answer this question.
> > In Section 3.1, when evaluating the simulated physical systems, the procedure of obtaining the ground-truth of the connectivity/interaction matrix is not clear, since the method is in an unsupervised manner. Why use a single interlayer to get the "ground truth" matrix? More details and explanations should be included.
>
> * First, we guess that your inquiry holds to section 4.1 rather than 3.1 since it explains the implementation of the simulated physical system.
> * If that's the case, then 'single inter-layer should indicate the one in the test example in Figure 5, where we test the models with the system with handcrafted adjacency matrix (Fig. 5A). Here, the term 'single inter-layer link' is not a 'layer' in a neural network, but an 'inter-layer link', which indicates the link connects two layers of system networks and also a common term in network science. We manually added this inter-layer link when we craft the adjacency matrix (and therefore the system trajectories) to test whether this weak but important link can be spotted by various models.
> * So, we don't use a single inter-layer link to get the ground-truth matrix, we deliberately set the inter-layer link in the adjacency matrix for this specific test case in Figure 5.
> * And, indeed, our method and model are trained in an unsupervised manner, so no ground-truth interaction strengths are needed before the training.
>
> We slightly modified the description in Section 4.1.2 and the caption of Figure 5 to emphasize that the adjacency matrix in Fig. 5(A) is manually crafted for testing and clarifying the whole situation. Please let us know if this explanation seems insufficient or if our interpretation of your inquiry was incorrect.
>
> ---
>
> 3. For CMU motion data, we use raw data instead of the difference to 'calculate the loss' because the scale of state differences in the data was quite small, and we found that the experiment with raw value converged faster. There are several ways to bypass this scaling problem, such as normalizing the data after calculating the differences, but we chose to use raw values instead. But, we found that the previous manuscript says 'model predicts the raw value', which is misleading, and hence we revised this description in the main manuscript.

---

> > ### Author Response · Authors · 2022-11-15
> > **Response to Reviewer cYy1 (2/2)**
> >
> > 4. As we mentioned in the main manuscript, a system of walking humans has no well-known dynamics with a continuous interaction strength, and therefore no clearly defined ground-truth interaction strength. And, it is hard to present a quantitative measure when there is no ground-truth value is available. Besides the argument like 'our inference has a higher chance to be correct since our model showed lower error', we stated a qualitative statement in the main manuscript about the inference result, where the inference of NRI shows virtually no internal structure, while RAIN+PA shows a more modular adjacency matrix when partitioned by the (physically) nearby nodes and hence more interpretable.
> >
> > ---
> >
> > 5. Indeed, transformer-based approaches suffer from $O(N^2)$ time complexity. If the system size $N$ is extremely large, $O(N^2)$ time complexity of the attention mechanism becomes a bottleneck. In our experiments, we found that the PA mechanism increases computation by 1.2%(N=5, spring), 6.5%(N=10, spring), and 34.5% (N=31, motion), compared to a vanilla RAIN (N>30 is a relatively huge setting in the field of relational learning). One promising resolution for this would be adopting some newly proposed attention mechanisms with linear time complexity and virtually no performance drop [1-2] to calculate PA. We added this discussion and references about reducing time complexity in the conclusion section of the main manuscript.
> >
> > ---
> >
> > 6. For minor mistakes:
> > In Figure 2, it's "+". It indicates the weighted sum in PA.
> > In Figure 5(D), we added the description in the caption that the trajectory predictions are on the $xy$-plane.
> > We checked the typos in References, especially some inconsistent capital letters in the journal name.
> >
> > ---
> >
> > We hope our revised manuscript and responses have helped you understand our work better, and we look forward to your positive feedback. Thank you!
> >
> > [1] Wang, S., Li, B. Z., Khabsa, M., Fang, H., & Ma, H. (2020). Linformer: Self-attention with linear complexity. arXiv preprint arXiv:2006.04768.
> > [2] Shen, Z., Zhang, M., Zhao, H., Yi, S., & Li, H. (2021). Efficient attention: Attention with linear complexities. In Proceedings of the IEEE/CVF winter conference on applications of computer vision (pp. 3531-3539).

---

### Official Review · Reviewer_zkVt · 2022-10-31

**Confidence:** 4
**Correctness:** 4
**Technical Novelty And Significance:** 3
**Empirical Novelty And Significance:** 3
**Recommendation:** 8

**Clarity, Quality, Novelty And Reproducibility:**

The writing is clear, novel, and seems to offer enough implementation detail to reproduce. But the sharing of the code would be even better.

**Strength And Weaknesses:**

Strength:

The triplet of (trajectory encoder + interaction graph + trajectory decoder) seem to work well, with the important added tricks such as modified RNN's pairwise attention (Eq 4-5), prior-less graph with transformer-like attention (Eq 7-8). Those methodologies seems sufficiently different from previous works and are novel.

Table 2 indicate a significant performance gain by using RAIN.

Weakness:

The paper draws similarity between the neural network and the physical dynamic system. However, the reviewer can't find any physical constraints are directly applied in the neural networks, nor at the neural network losses. The reviewer would like to see the actual dynamic system encoded in the NN architectures to generate more guaranteed realistic trajectories.

The experiment results include a simulated data (Table 2) and motion capture data of human walking (Table 3) seems toy-ish (although some previous works also uses those data). The reviewer would like to see how this method perform in a more complex scenario such as in https://waymo.com/open/

**Summary Of The Paper:**

This paper proposed a RAIN to infer interactions among agents. It learns the attentive weight and unknown system dynamics, which is also helpful to predict future trajectories. The author(s) verified that RAIN can infer the system dynamic parameters and interaction graphs, and can outperform baseline discrete models.


**Summary Of The Review:**

The main contribution of the paper is the proposal of the RAIN + PA which achieves a surprisingly good result in Table 2. Probably the contribution of adding PA to the RAIN is more important (empirically) than the main model (RAIN).

The introduction of PA is not unheard of, but is new in the interaction/prediction setting. This is the main reason it is marginally above the acceptance threshold.

The reason it is not getting higher ratings is because the performance of the main model RAIN without PA is somewhat disappointing. Also, the reviewer would like to see the method tested on some more non-trivial datasets to verify its validity.

---

> ### Author Response · Authors · 2022-11-15
> **Response to Reviewer zkVt**
>
> Thank you for your thorough and detailed review of the manuscript. We carefully checked the issues and revised our manuscript as follows. (We recommend you check the overall comment before reading this.)
>
> ---
>
> 1. As mentioned in the overall comment, we added a recent baseline, GATv2 (2021), to enrich our performance comparison and verified our model's superiorities over the new baseline. (will be ready before 11/17) Also, we separated the 'Related studies' section from the current introduction and added more recent references to help the reader's understanding of the field.
>
> ---
>
> 2. The final (camera-ready) version will have the link to the GitHub repository which contains codes for the module implementation and the simulation. We anonymized (basically, removed) it since the current version is the preprint for the double-blind review. Thank you for your understanding. (If it is allowed to reveal the repository during the rebuttal period, please let me know.)
>
> ---
>
> 3. As you pointed out, RAIN does not possess any form of constraint to the system. This, however, actually has its own merits and demerits in terms of model construction. If the given model or physical system is well specified and we already have prior knowledge of the system dynamics, one may incorporate such constraint directly to the NN architecture of RAIN and improve the performance. But, by deliberately removing such system-specific constraints, a model like RAIN can be applied to virtually any system, especially when the system dynamics are unknown, which is actually one of the key strengths of RAIN (as emphasized in the abstract and the main manuscript). In this work, we focused on constructing a more general framework that can handle vastly different systems, hence no specific constraints are added. We look forward to seeing follow-up studies that focus on a more specific system and fine-tune the RAIN with such constraints.
>
> ---
>
> 4. Different from many previous works, our main purpose of this work is not focused on better prediction (it's a byproduct), but rather, better inference on (weighted) interaction graphs. That's why the models used here are relatively toyish since we needed simple but fundamental models to perform a ground-truth comparison. Nonetheless, as you suggested, we also agree that demonstrating the capability of RAIN with more complex scenarios, such as pedestrian prediction, would be an interesting future topic.
>
> ---
>
> 5. We acknowledge that the vanilla version of RAIN (RAIN without PA) performs somewhat poorly in the Kuramoto system, although it outperforms every other baseline (including the new one) except RAIN + PA. Here, we would like to note that this demonstrates the hardship of the problem itself and clearly shows that changing the extraction algorithm only is not sufficient for properly predicting complex models in the interaction inference problem.
>
> ---
>
> 6. If you can tell us about previous works with similar concepts as PA in other domains or settings, we will gladly add them as references.
>
> ---
>
> We hope our revised manuscript and responses have helped you understand our work better, and we look forward to your positive feedback. Thank you!

---

> > ### Comment · Reviewer_zkVt · 2022-11-29
> > **Thanks for the clarification**
> >
> > Thanks for the clarification on the purpose of the paper being inferencing interaction graphs. The reviewer agrees that having toy-ish dataset is more straight-forward to evaluate and benchmark different algorithms.

---

### Official Review · Reviewer_5vb5 · 2022-11-03

**Confidence:** 2
**Correctness:** 3
**Technical Novelty And Significance:** 2
**Empirical Novelty And Significance:** 3
**Recommendation:** 6

**Clarity, Quality, Novelty And Reproducibility:**

**Clarity**

The paper generally lacks clarity, as mentioned in the section above. In addition to the aforementioned points, authors could consider including an algorithm block to summarize their method.

**Quality**

The results appear to be strong when compared to NRI, but it is unclear to me how strong and relevant of a comparison that is given NRI dates from 2018 and was designed for graphs with discrete interactions. I am also curious for the authors to describe how NRI would perform given access to a higher number (N > 4) of edge types. Is it possible, for instance, that NRI(16) outperforms RAIN+PA, but at a huge computational cost?

**Novelty**

Being unfamiliar with previous works in this field, it is difficult for me to assess novelty.

**Reproducibility**

If authors follow through with their claim of releasing data and models on Github, reproducibility should not be an issue.




**Strength And Weaknesses:**

**Strengths**

The main strength of the paper appears to be its empirical evaluation. Evaluating on three different very different kind of systems (plus a multi-level variant of the ball-spring system) provides a significant insight into the advantages of RAIN over NRI, which can only assign discrete interaction strengths to graph edges.

The results section is further strengthened by a varied analysis combining trajectory predictions, connectivity matrices and correlation coefficients. I found that this multi-modal approach helped my appreciation of the results despite being unfamiliar with previous works.

Finally, the problem of learning continuous interaction graphs from data seems challenging and highly relevant to both future research and applications.

**Weaknesses**

The first issue with the paper is the lack of a Background section clearly describing the problem and introducing the key variables and general approaches. Without a background section, the paper is hard to follow since it is unclear what assumptions can be made about the system or what constitutes success.

Secondly, I found the Section 2 lacked clarity, partially due to the issue above but also due to writing and inconsistent notation.
For instance, it is not clearly stated before the method whether one trajectory describes the evolution of the entire system, or whether it is one trajectory per node. Based on the section 3, I believe it to be the latter, but sentences like the following at first let me believe it was the former, when trying to understand the method:
"The encoder receives $T_{enc}$ steps of trajectories $x_1, x_2, ... , x_{T_{enc}}$ , each consisting of $R$ state variables of $N$ agents, and produces corresponding $T_{enc}$ hidden states, $h_1, h_2, ... , h_{T_{enc}}$."

Relatedly, the authors state "we calculate the attention between _same-time hidden states_ to assign weights to their contribution". The term "same-time hidden states" is never clearly defined, despite being key to PA. Additionally, Figure 2 clearly features a sum over different time-steps to compute the Pairwise Attention, which seems contradictory.

Another point of confusion is that variables used in figures 1 and 2 ($h'_{ij}, A_\theta, w_t$) are absent from the main body of the paper, including equations. Given the two previous points, I found Figures 1 and 2 to be of little help to my understanding, overall.

I also found it confusing that nodes (e.g. balls in the ball-spring system) be called "agents", given that they take no action. That said, feel free to ignore this comment if this is standard terminology in the field.

Finally, on the experimental side, results reported lacked confidence intervals or other measures of uncertainty. I also found it slightly odd that the only baseline reported dates from 2018.



**Summary Of The Paper:**

This paper introduces RAIN, a method to learn relational graphs between different nodes ("agents") of a system from time-series data describing the trajectories followed by said nodes.

RAIN is composed of a) an encoder computing an embedding of the trajectories seen so far, b) a graph extractor that infers the interaction strength between nodes. The authors also combine RAIN with a "pairwise attention" (PA) mechanism, which greatly improves the reported performance of RAIN.

The authors describe their method and then evaluate it on spring-ball models, Kuramoto oscillators and motion capture data of a person walking.

**Summary Of The Review:**

I believe the paper addresses a challenging and relevant problem, and proposes a novel architecture to do so. It also presents a range of strong empirical results, albeit one that raises a few questions about the reported baseline. Also, the paper suffers from a global lack of clarity and a missing background section, which is its main weakness in my opinion.

Given that, I do not believe the paper is ready for publication, but I am willing to reassess if the issues mentioned are resolved.

---

> ### Author Response · Authors · 2022-11-15
> **Response to Reviewer 5vb5**
>
> Thank you for your thorough and detailed review of the manuscript. We carefully checked the issues and revised our manuscript as follows. (We recommend you check the overall comment before reading this.)
>
> ---
>
> 1. As mentioned in the overall comment, we added a recent baseline, GATv2 (2021), to enrich our performance comparison and verified our model's superiorities over the new baseline. (will be ready before 11/17) Also, we separated the 'Related studies' section from the current introduction and added more recent references to help the reader's understanding of the field.
>
> ---
>
> 2. In the introduction and related studies section, we added more backgrounds such as the problem statement and variables in concern.
>
> ---
>
> 3. As for the issues related to sections 2 and 3, both statements you mention are correct for their own reasons. Normally, we regard a trajectory as a time evolution of a single agent, and it means each node (or agent) has its own trajectory. On the other hand, the LSTM encoder receives the states of all agents at a single timestep and repeatedly updates the hidden states through the entire timestep. This indicates that the LSTM encoder works parallelly in terms of agents, but sequentially in terms of time. Thus, the descriptions in section 3 are also technically correct.
>
> I presume that the main reason for the confusion comes from the insufficient descriptions of the notations, and the absence of distinguished notations between a single trajectory and an entire state of the system (which is highly related to your first claim, weak problem statement). Hence, we revised the main manuscript by adding more detailed descriptions and notations to clarify this issue.
>
> ---
>
> 4. Here, the term 'same-time' in PA indicates that the attention values are calculated between $h_i^t$ and $h_j^t$ ($K_{\text{pair}, i}^t$ and $Q_{\text{pair}, j}^t$, technically) only for same time t, not between every possible $t=0$ to $T$, as conventional full attention mechanism does. In other words, we do not perform a double sum over $t_n$ and $t_m$ $(0 \leq n, m \leq T)$ for each i and j, but only a single sum over $t$. (Indeed, the weighted sum after this gathers all such values from different timesteps to create a refined hidden state). We added this description to the main manuscript for clarity.
>
> ---
>
> 5. For unused variables in Figure 1 and Figure 2, we revised our figures to align those variables with the descriptions in the main manuscript. More precisely, we fixed $h_{ij}'$ into $h_{\text{dec}, i}$ and added a description about $A_\theta$ in Figure 1, and $w_t$ into $\alpha_{\text{pair}}^t$ in Figure 2.
>
> ---
>
> 6. To check the case where NRI has N > 4 edge types, we performed additional experiments and reported them in the supplementary material, section 5. Here, we used the Kuramoto model with 5 oscillators as a test system, and trained NRI with 8, 16, and 32 edges. We found that after NRI(8), no significant performance gain was observed, and verified that RAIN+PA outperforms all of the newly tested NRIs. Note that this takes significantly higher memory cost (almost linear to the number of the edge types) and computational time (not as drastic as memory cost, but one might need to reduce the batch size due to the memory limitation, which increases effective training time in general). Also, it was impossible to report the correlation of the best-performing case for these NRIs, since it needs an extremely high number of permutation tests (for NRI(32), we have $32! = 2.63 \times 10^{35}$ cases). Thus, even if NRI with more edge types shows lower prediction error, using it as an interaction strength inference model is nearly infeasible.
>
> ---
>
> We hope our revised manuscript and responses have helped you understand our work better, and we look forward to your positive feedback. Thank you!

---

> > ### Comment · Reviewer_5vb5 · 2022-11-19
> > **Reply**
> >
> > I thank the authors for their thorough response, and for taking the time to update the submission in light of the feedback received, which includes running an additional baseline. I have updated my score accordingly.
> >
> > For future submissions, I would recommend that the authors highlight the revisions they bring to the manuscript in a different font colour. This will make it easier to see changes and increase the potential effect of their rebuttal.

---

### Author Response · Authors · 2022-11-15
**Overall comment**

## [**NOTICE**]
The results from GATv2 are added, and the main manuscript is accordingly revised. Please check the ***'GATv2 Result'*** part of this comment and the revised manuscript.

---

**Overall comment for all reviewers**: First of all, we would like to express our heartfelt thanks to all reviewers who gave helpful reviews of our manuscript. Overall, we have corrected and added the followings:

---

1. All reviewers requested a comparison with the recently proposed baseline, and some reviewers even recommended us some models to test. However, the reason we did not include more baselines in the first place was that it was hard to find proper baselines. Many of the previous models are not capable of explicitly inferring the interaction strength or rely on the traditional Graph Attention (GAT, Veličković, P. et al, 2017) module, which we empirically checked its insufficiency for continuous weight inference. (In fact, this work started because GAT didn't work for our problem). Fortunately, during the rebuttal period, we found the advanced version of the GAT, namely, GATv2 (Brody, S. et al, 2021), which is analytically proven to be strictly more expressive than the original GAT. Hence, we added GATv2 into every experiment as a new baseline to enrich our performance comparison and again verified our model's superiority over one of the latest architectures.

## [***GATv2 Result***]


* In summary, for simulated systems, GATv2 showed performances on par with vanilla RAIN and even better performances in some categories (especially in prediction error), but RAIN with PA still kept its place in terms of both strength inference and prediction error (See Table 1 and 2).
* In the empirical system (motion data), GATv2 showed worse prediction error than NRI baselines (and both of our models, see Figure 6A).
* During the rebuttal period, we were quite impressed with the overall performance of GATv2, whereas the original GAT performed not even close to the vanilla RAIN. We envision that the advanced version of our model might be possible by wisely combining the best intuitions from both RAIN and GATv2.
---

2. We separated the 'Related studies' section from the current introduction and added more recent references (along with their characteristics and limitations) to help the reader's understanding of the field's recent advances. Also, in the introduction, we added a general problem statement and notations.

---

3. During the revision, we found some discrepancies between Figure 1 and the description of the main manuscript about the graph extraction module. The graph extraction module of RAIN uses a small MLP ($A_\theta$ in Figure 1) to calculate attentive values, whereas the previous manuscript described it as a transformer-like structure. (Actually, we also tested transformer-like structures before and found the current MLP version works better.) This is corrected in the revised version and now agrees with Figure 1 and the descriptions in the supplementary material.

---

4. We addressed various issues raised by each reviewer and provided answers to the questions, by performing extra experiments and adding detailed discussion to either the main manuscript or the supplementary material. (This can be checked in the comments for each reviewer.)

---

5. As an extra experiment, we tested $N=30$ cases (spring-ball system) with link probability $p=0.1$ to demonstrate the capability of RAIN in high-number settings. We reported the result in the supplementary experiments.

---

Some minor points:
* We added the experiment result of SingleLSTM of motion data to Figure 6(A), for completeness.
* We clarified the notations in Figure 1 and Figure 2 and their captions to align better with the model description section.
* Overall, we corrected typos, formatting issues, and unclear descriptions in the manuscript and supplementary material.

---

We think this revision process has made the paper clearer and stronger, and we would like to thank the reviewers again for providing quality reviews. During this rebuttal period, we added a new baseline, performed various supplementary experiments, and added more detailed descriptions to the text. We hope that this revised manuscript and responses will have a positive impact on the final evaluation. Thank you.

---

### Comment · Area_Chair_tr22 · 2022-11-24
**Reviewers, please engage in discussion**

Dear reviewers,
We are now approaching the end of the discussion period and so far nobody has engaged in discussion with the authors.
The authors both updated the paper and provided detailed responses to the reviews. Please reply, clarifying whether your concerns were addressed and if not, why not.
Also, if your concerns were indeed addressed either update your score or clearly state why you still believe the score is appropriate.

Many thanks for making this conference a success and for taking your role seriously.

AC

---

### Decision · Program_Chairs · 2023-01-20

**Decision:**

Accept: poster

**Justification For Why Not Higher Score:**

No theory or guarantees, some of the results miss confidence intervals.


**Justification For Why Not Lower Score:**

Intuitive method, strong empirical results. New baselines that got added during the rebuttal period and other improvements of the paper.


**Metareview: Summary, Strengths And Weaknesses:**

Summary:
The main contribution of this paper is the relational attentive inference network (RAIN) for inferring continuously weighted interaction graphs without any ground-truth interaction strengths. This is achieved using a novel pairwise attention mechanism and a graph transformer to extract heterogeneous interaction weights for each pair of agents. The RAIN model is shown to accurately infer continuous interaction strengths in simulated physical systems in an unsupervised manner. Additionally, the model is able to predict trajectories from motion capture data with an interpretable interaction graph.

Strength:
Intuitive method, strong empirical results. New baselines that got added during the rebuttal period and other improvements of the paper.

Weakness:
No theory or guarantees, some of the results miss confidence intervals (e.g. Table 2). This should be addressed for the CRC.


**Note From Pc:**

if the above contains the word "oral" or "spotlight" please see: "oral" presentation means -> notable-top-5% and "spotlight" means -> notable-top-25%. As stated in our emails, we are disassociating presentation type from AC recommendations